# UAV Detection Using Reinforcement Learning

**DOI:** 10.3390/s24061870

**Published:** 2024-03-14

**Authors:** Arwa AlKhonaini, Tarek Sheltami, Ashraf Mahmoud, Muhammad Imam

**Affiliations:** 1Computer Engineering Department, Interdisciplinary Research Center of Smart Mobility and Logistics, King Fahd University of Petroleum and Minerals, Dhahran 31261, Saudi Arabia; amalkhonaini@iau.edu.sa (A.A.); tarek@kfupm.edu.sa (T.S.); ashraf@kfupm.edu.sa (A.M.); 2Computing Department, Applied College, Imam Abdulrahman Bin Faisal University, Dammam 34212, Saudi Arabia; 3Computer Engineering Department, Interdisciplinary Research Center for Intelligent Secure Systems, King Fahd University of Petroleum and Minerals, Dhahran 31261, Saudi Arabia

**Keywords:** radio frequency, Unmanned Aerial Vehicles, hierarchical reinforcement learning, detection and identification, REINFORCE

## Abstract

Unmanned Aerial Vehicles (UAVs) have gained significant popularity in both military and civilian applications due to their cost-effectiveness and flexibility. However, the increased utilization of UAVs raises concerns about the risk of illegal data gathering and potential criminal use. As a result, the accurate detection and identification of intruding UAVs has emerged as a critical research concern. Many algorithms have shown their effectiveness in detecting different objects through different approaches, including radio frequency (RF), computer vision (visual), and sound-based detection. This article proposes a novel approach for detecting and identifying intruding UAVs based on their RF signals by using a hierarchical reinforcement learning technique. We train a UAV agent hierarchically with multiple policies using the REINFORCE algorithm with entropy regularization term to improve the overall accuracy. The research focuses on utilizing extracted features from RF signals to detect intruding UAVs, which contributes to the field of reinforcement learning by investigating a less-explored UAV detection approach. Through extensive evaluation, our findings show the remarkable results of the proposed approach in achieving accurate RF-based detection and identification, with an outstanding detection accuracy of 99.7%. Additionally, our approach demonstrates improved cumulative return performance and reduced loss. The obtained results highlight the effectiveness of the proposed solution in enhancing UAV security and surveillance while advancing the field of UAV detection.

## 1. Introduction

Unmanned Aerial Vehicles (UAVs), commonly known as drones, have witnessed rapid technological advancement in recent years. By freely navigating the airspace, UAVs have the capability to be utilized in a diverse range of applications, including aerial surveillance in military operations [1,2], law enforcement tasks [2], as well as providing delivery services within residential areas [3,4]. At first, UAVs were developed to serve military-based applications. Later, the quality of civilian services was enhanced by using UAVs for simple tasks [5]. In recent years, smart cities have gained huge popularity where they prompt innovative projects, research, and ideas for a “smart” city that is environmentally and socially sustainable [6]. Smart cities are using UAVs to participate in emergency situations [7,8,9], surveillance [10,11], and deliver goods/medicine [12,13]. They have varying degrees of autonomy, including total autonomy, but are usually controlled by a ground-based human controller. Primarily, no matter where they are used, UAVs are a cost-effective option that are utilized to speed up processes, increase their flexibility, and improve their accuracy [14]. Moreover, UAVs have the potential to fly in hazardous conditions that humans would not be able to survive while upholding environmental protection [15,16,17,18]. Having built-in hardware, such as cameras or sensors, enables UAVs to achieve the assigned tasks successfully [19].

Apart from the significant benefits of UAVs, the likelihood of this incredible technology being misused increases in direct proportion to its popularity. UAVs’ increasing presence could be used to intrude on others, obtain data illegally, or gain unauthorized access to private establishments. Hence, they raise concerns regarding privacy, security, and safety. Consequently, the advancement of reliable and effective techniques for UAV detection and identification has become a pressing need.

Machine learning (ML) techniques created a great impact that serves the task of UAV detection and identification. Reinforcement learning (RL) is a subset of ML algorithms that investigates how agents should behave in a particular situation to achieve a certain goal or obtain the maximum rewards [19]. Effective traditional UAV detection approaches include radar systems [20,21], visual surveillance [22,23], radio frequency (RF) signals [24,25,26], and acoustic sensors [27]. According to [28], radio waves are used by radar systems to identify objects in the airspace. Visual detection uses image processing systems to detect and track UAVs based on their airframe features and overall appearance. UAVs can also be detected by analyzing their RF signals in the communication medium. Acoustic sensors depend on identifying the unique sound signatures of UAVs to confirm their presence.

Due to the advancement of UAVs, the detection process has become more difficult over the years. Modern UAVs are more flexible, reduced in size, and they embrace autonomous behavior; this gives them the ability to operate in small spaces and avoid detection by conventional means. Visual and acoustic-based detection methods are expensive and can be limited in a variety of ways, including utilizing stealth technology, modifying the rotors and physical form of the UAV, using low-noise rotors, and generating natural noises, such as white noise [29]. Innovative detection techniques are needed to overcome the limitations of these approaches and address the evolving threat of UAVs. In this article, we propose a novel approach for detecting and identifying intruding UAVs based on their RF signals using hierarchical RL (HRL).

Using RF sensing units not only lets us detect the existence of intruding UAVs but also helps identify the UAV’s manufacturing model and gain insights into its control mode. Hence, in the context of UAV detection and identification, RF signals provide a rich source of information that can be used to distinguish UAVs from other flying objects in the environment [30]. RF signals have a great advantage over some other detection techniques in which they can operate in diverse environmental climate conditions. Unlike visual detection, RF signal detection is not affected by factors such as lighting conditions, also it can penetrate through various materials such as walls allowing for the detection of UAVs even in obstructed environments. Therefore, RF sensing-based methods for UAV detection and identification are sufficient for real life applications [31]. That is why academics like [30] were inspired to create droneRF [32], an open-source database of RF signals from various UAV models in various flying modes.

Despite the growing need to detect and identify intruding UAVs, there is a noticeable lack of studies that focus on using RF signals and RL techniques for this purpose. To the best of our knowledge, there is no existing research that proposes an RF-based UAV detection and identification system using RL, creating a significant research gap in the field. This motivates us to explore and implement a novel framework that leverages HRL to detect and identify UAVs based on their RF signals. As the fundamental technique, we use HRL, which relies on the policy gradient method based on the REINFORCE algorithm. We designed an adaptable data-driven strategy that can successfully detect and identify UAVs in real-time scenarios.

### Article Contribution

The primary contribution of this research is to provide an HRL-based solution for UAV detection and identification using RF signal data. The proposed approach consists of training the UAV agent, building its experience, and making informed decisions based on captured RF signals. This process enables effective differentiation between UAVs and other flying objects effectively. Furthermore, the hierarchical nature of our approach allows for the decomposition of the problem, which increases the accuracy compared to non-hierarchical solutions.

The remainder of this article is organized as follows: Section 2 provides an overview of the previous literature work focusing on different detection techniques in the fields of ML and RL. Section 3 presents a comprehensive insight into our methodology, including the system model and dataset analysis. Our hierarchical detection and identification system is proposed in Section 4. Finally, Section 5 presents the evaluation of the proposed solution throughout both the training and testing stages, as well as a full analysis of the results.

## 2. Literature Review

ML is proven to be incredibly effective in a variety of sectors when hand-coded solutions fail, and massive volumes of labeled data can be acquired [26]. In the case of UAV detection, ML algorithms were utilized through various detection methodologies including the analysis of RF signals [33,34,35], sound characteristics [27,36,37,38], and visual cues [39,40,41]. There is a great interest in applying RL techniques to help in the detection and identification of intruding UAVs. In this section, we go through some state-of-the-art solutions that have been used to detect intruding UAVs using a range of ML and RL detection approaches.

### 2.1. Radio Frequency Detection Techniques Based on Machine Learning

Micro UAVs (MUAVs) have become pervasively popular for different purposes, especially monitoring purposes. However, to avoid undesired surveillance, an advanced MUAV detection approach [33] is proposed. This approach applies feature extraction and reconstruction using artificial neural networks (ANN). They removed the noise in RF signals by using the background estimation approach. Subsequently, by using the singular value decomposition (SVD), the primary components of the RF signals are extracted, which helps in reconstructing the effective RF signals. The higher-order cumulant (HOC) algorithm enhances the aignal-to-noise ratio (SNR) of RF signals, which possibly reduces Gaussian noise to zero. ANN utilizes other characteristics including standard deviation (STD) to achieve a higher accuracy percentage, which is more than 80% for different positions of the target. Their proposed solution outperformed other traditional well-known algorithms [42,43].

A passive RF surveillance system is proposed [34] to detect and classify UAVs successfully by distinguishing between the signals coming from background noise, UAV controller, and wireless signal interference. They use a Naive Bayes decision technique that is based on Markov models to detect RF signals, which are fed to an ML-based classification and identification system to further categorize the model of the UAV. They apply the neighborhood component analysis (NCA) algorithm to extract the most significant features in the RF signal. Additionally, the system was evaluated in different environmental settings including different SNR values where they found that under 25 dB SNR, the k-nearest neighbor (kNN) algorithm scored 98.13% accuracy.

Another research for MUAV detection is proposed in [35] using RF fingerprints of the transmitted signals between the controller and MUAV. There are two phases: detection and classification. First, raw signals are divided to frames and converted into the wavelet domain in the detection phase to limit the amount of data that has to be processed and eliminate bias from the signals. To determine the existence of UAV in every frame, a Naive Bayes method is used by creating independent Markov models for classes of UAVs and non-UAVs. They trained their work using kNN, neural networks (NN), discriminant analysis (DA), support vector machine (SVM), and random forest (RF). Their method leverages the energy transient signal in the classification phase, unlike conventional methods that only use time-domain signals and related features. This method can handle various modulation schemes and is more resistant to noise. The raw signals are analyzed to create the normalized energy trajectory using the energy–time–frequency distribution. Then, the energy transient’s start and end locations are identified by searching for the most significant deviations from the average of the energy trajectory. The energy transient is then used to obtain a collection of statistical features and the NCA algorithm analyzes the relevance of those features. Finally, those features are fed to a group of ML algorithms for classification. Their work was tested using a 100 RF signals database. The signals in the database came from 14 distinct UAV controllers as well as different SNR levels. The kNN algorithm accurately spotted all the MUAVs with an average accuracy of 96.3%.

Locating UAV controllers autonomously via generated RF signals is proposed in [26]. Signal spectrum analysis is carried out using an RF sensor array. With the sensor’s output, a convolutional neural network (CNN) is trained to be able to anticipate the orientation of the UAV controller. If at least two of these sensors are at an acceptable distance apart, the position of the controllers may then be determined from these directions. The mean absolute error (MAE) of their model is calculated to be 3.67°, which means a moderate positional inaccuracy of 40 m at a range of 500 m.

A sophisticated UAV detection and identification algorithm was proposed by [30]. They developed an open-source RF signals database called droneRF [32] that contains a variety of UAV models and modes, which was used to train their algorithm. Three deep neural networks (DNN) were used to detect the existence of UAV, the existence of UAV and the making model, and finally the existence of UAV, the making model, and the flying mode. Every DNN’s performance is tested with a 10-fold cross-validation technique along with several metrics The average accuracy for the first DNN (2 classes) is 99.7%. However, using the second DNN (4 classes), the accuracy is 84.5%, and finally, the third DNN (10 classes) scored 46.8%. Increasing the number of classes had an immediate negative impact on the classification results where they reveal a general reduction in performance, which can be attributed to the similarities between Bebop and AR UAV models in terms of the RF communication signals. Their results approve the viability of the created RF database for UAV detection purposes.

An efficient UAV detection and identification algorithm using ML was presented in [24]. Their work is based on ensemble learning classifiers where four classifiers are working in a hierarchical fashion. To enhance the output of the detection technique, RF signals go through filtering feature extraction and pre-processing. To start, the detected sample goes through the first classifier to determine the presence of the UAV. After that, the second classifier identifies the making model of the UAV. The last two classifiers are working with AR and Bebop samples to determine the flying mode. Their work outperforms existing detection algorithms when tested using the same dronRF dataset [32]. With an accuracy of 99.2%, their work can examine whether a UAV is flying in the environment, and it can immediately identify the making model of the UAV and its flying mode.

Also, ref. [44] proposed another hierarchical learning framework for reliable RF-based UAV detection and identification. First, signal detection was conducted using the stacked denoising autoencoder (SDAE) and local outlier factor (LOF) algorithms. SDAE compressed the RF signals to minimize the space complexity while LOF is utilized to identify UAV signals through anomaly detection among Wi-Fi and Bluetooth transmissions. To identify the UAV, features were extracted from the UAV signals using the Hilbert–Huang transform (HHT) and wavelet packet transform (WPT) algorithms. XGBoost algorithm was the base learning technique that trained a three-level hierarchical classifier using the extracted features. The goal of the hierarchical classifier was to identify the UAV signal and determine its model and flying mode. They used the Cardinal RF (CardRF) dataset [45], created by capturing RF signals of UAV, UAV controller, Bluetooth, and Wi-Fi devices in an outdoor environment. They made CardRF an open-source dataset to enable future research in UAV detection and identification. The proposed SDAE-LOF framework’s performance was examined, and it achieved a detection accuracy of 89.5% for both UAV and non-UAV signals. Table 1 provides a clear summary of the reviewed RF detection techniques using ML techniques.

### 2.2. Detection Techniques Based on Reinforcement Learning

This section provides a comprehensive review of various detection techniques that use different reinforcement learning techniques and demonstrate their effectiveness.

#### 2.2.1. Detection Based on Acoustic and Visual Cues

Visual and sound cues form a fundamental part of detection techniques, enabling the accurate identification of objects. An integration of the deep learning model called YOLOv5 and a policy gradient RL technique was proposed to detect and track intruding UAVs by launching a chaser UAV within a confined area [46]. Their UAV agent is equipped with a monocular camera installed in the front. They integrated computer vision strategies along with a policy learning approach to train their system to control the policy and track intruding UAVs successfully. When the UAV agent catches an intruding UAV through its camera, the raw images are processed using YOLOv5, and the intruding UAV’s coordinates are fed to the RL control policy for the tracking mission. By training their system with a dataset that contains 20,000 images of drones with different orientations, they achieved a great detection accuracy of 97%.

An alternative proposal is the integration of the real-time object detector EfficientNet-B0 and the double deep Q-network (DDQN) method in [47]. The UAV agent uses a camera that is mounted on top to capture photos that are used to detect intruding UAVs by applying image processing. Each state consists of the processed image as well as a group of scalar values that determine the velocity and a variety of distance evaluations. After training the DDQN model, they employed a human operator to control their UAV agent for capturing stationary and non-stationary intruding UAVs. The performance results of both training and testing demonstrate the agent’s capability to capture intruding UAVs. Furthermore, the agent avoids colliding with any environmental obstacles with a success rate of at least 94%. Furthermore, a comparison was made between the performance of their DDQN model and that of human pilots. It was observed that the agent trained with DDQN demonstrated a faster process of capturing intruding UAVs. This is because human pilots find it difficult to fly and control the UAV using a remote controller. However, the agent using the DDQN model almost never misses the target while attempting to capture it.

As an alternative to the conventional, ineffective detection methods employed in the military, the military vehicle object detection method (MVODM) was proposed in [23]. Hierarchical feature representation and RL localization techniques form the foundation of this approach. The first strategy generates both the initial bounding box and the hierarchical feature representation. To enhance performance and accuracy, their strategy tends to represent the features on different levels. The second strategy employs RL to enhance the detection of the object by selecting the proper feature representation layer among the different levels produced earlier. To start the feature extraction algorithm, the image is inserted into the ResNet50 network. After that, to build the original bounding box, the captured features are passed into the following network using the hierarchical feature representation strategy. The final detection results are obtained by feeding the original bounding box and the hierarchical features into the refined localization strategy that is based on the RL technique. They evaluated the performance and effectiveness of their work by comparing it to different existing mainstream object detection algorithms including single-shot detector (SSD), region-based fully convolutional network (R-FCN), and YOLOv3. They created a dataset and used it for training and testing purposes where they included different classes of military vehicles. MVODM achieved the highest detection accuracy 86.7% for different types of vehicle classes.

An active RL-based object detection model was proposed by [22]. Their model can detect targets that were previously learned by their system. Instead of the traditional sliding window technique that consumes more time than needed, their approach uses a top-down search methodology that begins by examining the entire picture as input before focusing on localizing the precise locations of the target. The initial bounding box covers the whole image then it shrinks to a smaller bounding box by performing a series of transformations. They apply a dynamic attention-action strategy where the focus of the model will be on the content of the bounding box only. The agent chooses the next optimal action by analyzing the content of the region to determine the order of transformations. Each step taken by the agent should remove as much background as possible while maintaining the target inside the bounding box. Their model detects objects by processing 11 to 25 bounding boxes only.

To decrease localization error and energy consumption rate, passive acoustic localization based on RL for hidden mobile node (RMHPL) is proposed [48]. Underwater sites introduce a range of difficulties, including non-line-of-sight transmissions, drifting clocks, extended propagation time delay, and restricted bandwidth. These difficulties influence distance estimation errors, which lower the localization accuracy. The primary component of the network is underwater acoustic sensor networks (UASN), for data collection, transmission, and detection. to save energy and maintain their secrecy, hidden nodes will only locate themselves using received signals from anchor nodes. There are several anchor nodes established within UASN in the sea. All anchor nodes have a global positioning system (GPS) service, allowing them to locate themselves. The hidden nodes receive information about the current state, including the most recent time frame, localization error, and energy use, through RMHPL. Using this information, coupled with the Q-function, the hidden node can localize itself by having a better estimate of the time frame. When compared to previous methods, their technique minimized localization inaccuracy by up to 74% and energy usage by 85.6%. Table 2 summarizes visual and acoustic detection using RL techniques.

#### 2.2.2. Detection Based on Source Radiation

Radiation source detection is a critical nuclear technology dedicated to discovering anomalous radiation sources in a nuclear environment. In [49], a data-driven method using a double Q-learning technique based on convolutional neural networks (CNN) for nuclear radiation source detection autonomously is proposed. They formulated the detection task as a finite discrete Markov decision process (MDP). Their simulation scenario depicts an agent called ‘the detector’ that sweeps the perimeter of the restricted region to locate the source as quickly as feasible. Moreover, the agent will determine whether the radiation is coming from a background or anomalous source. They performed multiple tests to confirm the effectiveness of their work. First, in the radiation search test, they noticed that the Q-learning strategy required less search time and had a higher success rate than the gradient search algorithm. In the detector trapping test, the gradient search technique proved more prone to the detector trapping problem than the Q-learning strategy. The radiation source estimation test was used as the final test. The Q-learning approach scored 44% less searching time than the other methods.

In [50], a two-tier approach that consists of exploration and localization was proposed. The former seeks to efficiently gather information, while the latter uses the information to determine the quickest path to the source. They implemented a hierarchical control (HC) policy that helps in detecting the radiation source in unknown environment by going through two stages. First, the high-level controller will decide whether to execute an exploration or localization task. Then the low-level controller is responsible for executing the selected task by applying deep RL. Their agent ‘the detector’ is initially placed in a blocked area where the environment and exact geometry are unknown. The detector investigates the surroundings to build a clear perception of the environment. The detector should move around the detected obstacles to get out of the blocked area. Then, it should start looking for anomalous radiation sources by gathering radioactive source emissions and geometrical data about the surrounding area to help in moving toward the source of the radiation. They tested their proposed HC policy against four autonomous decision policies where HC scored the highest performance among all of them.

Furthermore, the ability of deep RL to make sequential decisions in radiation source detection was investigated in [51]. They proposed a novel neural network architecture that uses the idea of an actor–critic framework (RAD-A2C) to help in localizing the source of radiation. Their framework can locate many sorts of radiation sources including static or moving shielded nuclear sources. It also accounts for geographically variable background radiation rates using an attenuation model for distinct environment substances. Also, it can detect multiple sources of radiation at once. RAD-A2C uses particle filter gated recurrent unit (PFGRU). At each timestep, the PFGRU generates a geographical prediction, which is combined with the observation and propagated into the actor–critic framework (A2C). The actor layer determines actions based on the observations recorded by the PFGRU module. The critic layer determines the expected reward of the agent. They tested their performance in localizing the source of radiation in an environment that is subjected to a variety of SNR values. For their tests, they used [52,53] as an inspiration to create a hybrid controller RID-FIM after that they compared their results against it and against the gradient search algorithm. The most significant aspect of radiation source search is episode completion. In a high SNR environment, the gradient search algorithm had the greatest performance but when reducing SNR, the performance decreased dramatically. In a low SNR setting, the RID-FIM showed a slightly greater chance of failing to complete episodes. RAD-A2C was the only formulation that scored more than 95% of episode completion on both low and high SNR. Table 3 summarizes the reviewed radiation source detection techniques in the source radiation field in a 2D environment.

#### 2.2.3. Detection Based on Radio Frequency Signals

RF-based detection is a promising approach that plays a crucial role in accurately identifying objects by analyzing the RF signals in the communication medium. UAV dynamic radar networks (DRN) and RF sensors were utilized to facilitate target detection and mapping [21]. To enable UAV agents to detect objects with the greatest degree of accuracy and to improve their map reconstruction, they developed a multi-agent RL framework. UAV actions are based on knowledge gained from the DRN, providing the agents with a broader perspective of their surroundings, and enabling them to thrive which improves the network’s overall performance. The agents in the scenario are either working together by communicating through the network or working independently. In their scenario, targets are only those who are exchanging beacons that are detectable using RF sensors. Each UAV can sense the surroundings, estimate the state, allocate the needed resources (in case of multiple targets), and estimate the policy. When a UAV is working independently, it uses MDP for detection and Q-learning for navigation. When working within a group the UAVs use Q-sharing. During the evaluation, it was observed that including the agent in a group led to successful completion of the detection and navigation mission in a reduced amount of time. On the other hand, independent UAV detection and navigation are not as successful.

Moreover, the progressing reward policy gradient (PRPG) [25] is proposed as a deep RL-based algorithm to improve the accuracy of detecting three-dimensional (3D) objects using reflected RF signals. PRPG incorporates their agent, a metasurface within a scenario that consists of an omnidirectional antenna and a directional antenna in a 3D environment containing the targeted object. The metasurface, as described by [54], is a synthetic thin layer of evenly distributed reconfigurable components in a two-dimensional array. It can change the reflected RF signals and conduct beamforming by modifying the configuration of each reconfigurable component. PRPG primarily focuses on maximizing the accuracy of sensing by reducing the cross-entropy loss. First, the agent selects an action based on neural networks. Then, they used a two-fold training approach to enhance the policy and reduce cross-entropy loss. The performance of the PRPG algorithm is tested against different ML algorithms, and PRPG scored the lowest cross-entropy loss.

RF detection could also happen in an indoor search and rescue (SAR) mission, ref. [55] proposed to employ RL technologies to facilitate identifying targets by detecting RF signals transmitted by their mobile phones. GPS is known not to function properly in indoor environments; thus, it was replaced with the received signal strength (RSS), which measures the attenuation of RF signals during the transmission and is based on the distance from the transmitter to the directional antenna of the UAV. Their reward was calculated based on the difference between the RSS measurements corresponding to the adjacent UAV locations. They created situations in which UAVs launch at a predetermined location. The UAV detects RSS levels in each location and acts based on an epsilon-greedy strategy. While moving from one location to the other, the reward will be calculated, and the UAV will be rewarded if it is heading in the right direction toward the target. They used an untrained Q-learning framework to work with the gathered RSS values, create a Q-table, and update it with each move, resulting in more rapid navigation to the target’s location. Their work succeeded in reaching the target in a reduced amount of time compared to another study that works with an omnidirectional antenna.

Two RL techniques were implemented to localize and track moving targets in [56]. Their focus is to localize different mobile targets by equipping their agents with a sensing unit to capture the different RF signal configurations. The state of the environment consists of information gained from the sensing unit along with the RF source of the target. Agents update their policy based on the ‘belief’, which is the probability distribution over all possible true states. Target localization and navigation tasks are modeled as partially observable Markov decision process (PODMP). They implemented two RL techniques to solve the problem of PODMP; the first one is Monte Carlo tree search (MCTS) while the second one is deep Q networks (DQN). While the two techniques performed similarly in terms of target localization, DQN showed robustness to more test scenarios. In terms of inference speed, MCTS showed slower performance than DQN because MCTS depends on the depth of the search tree and the reward function for each action. The reward function slows down the performance as it requires the ‘belief’ to go through a particle filter to update the policy. DQN is faster in the testing phase, but it takes significant time while training. Both models offer great advantages based on the context of the environment.

In response to the increasing demand for accurate occupancy detection in indoor environments for various applications such as energy management and security, a novel method for indoor occupancy detection using Bluetooth Low Energy (BLE) signals and RL techniques is proposed [57]. They studied the limitations of traditional methods that rely on motion sensors or Wi-Fi signals which may not provide precise and reliable results. To overcome these limitations, they proposed a DQN framework that uses RF features collected from BLE devices to optimize the detection of human presence which is known as occupancy detection. The system architecture consists of a BLE signal receiver, an agent, and an occupancy classifier. In training, they enhance the learning of the agent based on RF parameters for an optimized decision-making process. They evaluated the performance of their work by conducting several indoor and real-world experiments. They compared the results with existing methods, including Wi-Fi and motion sensor-based approaches. The experimental results demonstrate that the proposed BLE-based method exhibits greater performance with an F1 score of 89.2% compared to 86.5% of traditional methods. Therefore, the BLE approach offers potential benefits for various applications requiring accurate occupancy detection in indoor environments.

The lack of existing research studies, to the best of our knowledge, on intruding UAV detection and identification systems using RF signals and RL techniques highlights a significant research gap in this field. This gap presents a great opportunity for our article to make a valuable contribution. The primary objective of our study is to propose a novel system that effectively utilizes the unique features of RF signals within an HRL framework for accurate UAV detection and identification process. By addressing this research gap, we aim to advance the understanding and practical implementation of UAV detection and identification systems using extracted RF features, thus making a notable contribution to the field of RL. Table 4 provides a summary of the reviewed work in the RF signal detection field in a 3D environment.

## 3. Methodology

This section provides a comprehensive overview of the system model used for creating the droneRF dataset [32] and evaluates its viability for usage in our proposed UAV detection and identification system. First, we start introducing the system model and summarizing its contributing components. Next, we elaborate on the dataset structure, including the total number of gathered segments and samples.

### 3.1. System Model

On-board transmitters are commonly integrated into UAVs to control and operate them solely through RF signals. These transmitters play a crucial role in enabling the communication between the intruding UAV and the flight controlling unit. Typically, UAVs operate within the 2.4 GHz Industrial, Scientific, and Medical (ISM) band, a frequency range commonly used for various applications, including a wireless communication system that is also designated as an unlicensed spectrum, meaning that no specific license or authorization is required to use this frequency range. UAVs can be detected and identified from considerable distances by considering both the UAV’s characteristics and the receiver’s attributes within the surrounding environment. Additionally, the positioning of the controller used for transmitting the signals can be strategically arranged to facilitate human involvement.

Figure 1 illustrates the system model of the system. There are three components in the detection system: a UAV device, a flight controller to control the UAV, and an RF receiver used to find and identify the source signal by intercepting the active communications between the UAV and the flight controller. In droneRF [32], RF data for various types of UAVs are collected to produce a large informative dataset for UAVs’ RF signals. The dataset is built using three UAV types: Parrot Bebop, Parrot AR Drone, and DJI Phantom 3. Due to their different specifications, sizes, technologies, and prices, these three UAVs are common in different kinds of applications. Table 5 gives a clear description of each UAV type along with its specifications and properties.

The flight controller serves as a controlling unit, functioning with a specific application designed for its intended purpose. The primary objective of this application is to facilitate the transmission and reception of the main tasks, or modes, to the UAV through RF signals. Each UAV is equipped with a distinct application tailored to its specific tasks.

The RF receiver unit is responsible for capturing the data transmitted by UAVs in the form of RF signals through wireless communication. The receiver unit comprises two components: the first component records RF signals at the lower end of the spectrum, while the second one captures RF signals at the higher end of the spectrum. As stated before, RF signals act as the link of data exchange between the UAV and the flight controlling unit. Thus, commands and telemetry data can be conveyed through that link to control the UAV’s flight operations. Table 6 outlines the specifications of the USRP-2943 40 MHz RF receiver unit [58] used to collect RF signals. In droneRF [32], the RF signals are obtained, processed, and stored via a personal computer (PC) or any device that has LabVIEW software version 5 installed.

### 3.2. Dataset

The proposed approach uses droneRF [32], a public dataset for the detection and identification of different kinds of intruding UAVs. The previous section discussed the system model used for the experiments, providing insights into the different UAV models, flight controlling devices, and RF receiving units. This system model is essential as it serves as the foundation for the proposed approach. Unlike other forms of datasets that rely on simulations or laboratory experiments, the dataset itself is constructed based on real experiments, making it highly valuable for training and testing purposes.

This comprehensive dataset facilitates the detection of UAVs in the environment. It includes a wide range of RF activities, consisting of both RF background activities (representing situations when UAVs are not present) and RF drone activities (corresponding to scenarios with UAVs). Moreover, the dataset covers RF signals emitted by various UAVs operating in different flight modes. Many features inside these RF signals can be extracted and used as unique identifiers including the frequency patterns, modulation techniques, signal strength, and other relevant parameters. This enables not only UAV identification but also the determination of the flight mode of intruding UAVs, such as hovering, flying, and recording videos. Within the dataset, there are recorded segments available in their raw format for each class, obtained through multiple experiments. In terms of specific durations, the RF background activities are recorded for approximately 10.25 s, while the RF communication for each flight mode of the UAV has a duration of roughly 5.25 s. To conserve memory, the RF signals are saved as segments in a standard CSV file format. The recorded segments were categorized into three levels, resulting in a total of 227 segments, each of $20\times 10^{6}$ samples. Figure 2 and  Figure 3 provide a visual representation of the number of segments of each level of the experiments. The three levels in the dataset are listed below:**Level 1**: This level determines if a UAV is present in the environment or not, resulting in two classes:-Class one: No UAV;-Class two: UAV.**Level 2**: This level identifies the making model of the UAV that is detected in the first level. We categorize them into three classes, named after their models:-Class one: Parrot Bebop;-Class two: Parrot AR;-Class three: Phantom 3.**Level 3**: This level determines the flying mode of the detected UAV identified in Level 2. Parrot AR and Parrot Bebop have four flying modes resulting in four classes for each model:-Class one: ON (mode 1);-Class two: Hovering (mode 2);-Class three: Flying (mode 3);-Class four: Recording video (mode 4).

Based on the dataset’s preceding description [32], a hierarchical approach can be used to solve the detection and identification challenge by treating it as a multi-class classification problem. Previous research proposed a state-of-the-art approach that has shown great accuracy results in detection and identification. They constructed a hierarchical detection and identification system using ensemble learning with decision trees [24]. Our system implements the same hierarchy using four classifiers to determine the existence of the UAV, its model, and how it is being operated.

To improve the learning process and enhance performance, ref. [24] successfully reduced the computing complexity and enhanced the detection accuracy by removing bias in the captured signals by eliminating zero frequency components and applying the Fast Fourier Transform (FFT) to zero-mean signals in MATLAB 2022b. Figure 4 illustrates the normalized RF signals of two segments recorded in the experiment using two receivers. The RF activity of segment 1, represented in Figure 4a shows a typical behavior (no UAV), but for segment 3 in Figure 4b it indicates the presence of a UAV specifically a flying AR model. After reducing the noise and eliminating the bias, Figure 5 shows the different levels of the experiment highlighting the spectrum of each data class in the dataset while Figure 6 shows the data distribution of the three experiment levels.

## 4. Proposed Solution

This section presents our proposed HRL (Hierarchical Reinforcement Learning) approach by focusing on the detection and identification of UAVs using a policy gradient algorithm based on REINFORCE with entropy regularization. Our approach incorporates the advantages of RL techniques with RF unique features to learn optimal decision-making policies in binary and multi-class classification settings. First, we formulated the problem in question by defining the model of the system and the dataset. After that, we apply filtering and smoothing techniques to the dataset to enhance the quality of the input features. Finally, we introduce our HRL approach to solve the problem of multi-class classification hierarchically by using a UAV agent with multiple policies.

### 4.1. Data Pre-Processing

The droneRF dataset [32] contains RF data collected in their raw format using onboard sensing units. Even though [24] has reduced the noise and eliminated the bias, the RF signals still require pre-processing before being used in the learning process. This stage involves performing a filtering process to further remove noise and interference. Additionally, we employ methods to enhance data quality and extract meaningful features. The process consists of two components: data and feature engineering. Both components play a crucial role in shaping and refining the data for additional analysis. Data engineering focuses on converting the raw RF data into a suitable format using techniques such as signal processing and encoding categorical variables. It involves collecting, cleaning, and integrating data from diverse sources. On the other hand, feature engineering involves refining the output of data engineering to create the desired features for the learning process. By transforming raw data into more meaningful features that capture relevant information, we aim to provide our HRL framework more informative and representative input space.

As part of the preprocessing pipeline, we introduce a filtering stage for smoothing and noise reduction. This stage employs a uniform filter with a predefined window size, utilizing an averaging concept. The adjacent samples within the window are added together and then divided by the size of the window to calculate the average value. This filtering process effectively reduces high-frequency variations and enhances the data quality, resulting in smoothed input samples.

Both smoothed input samples and the engineered features are utilized within the HRL framework to learn optimal decision-making policies and take actions based on the given target values. By incorporating the smoothed input samples and engineered features, our model benefits from improved data quality, meaningful representations, and enhanced predictability, thereby enabling more effective decision-making in our scenario.

### 4.2. Hierarchical Reinforcement Learning Approach

The intruding UAV problem creates a great security risk for the whole world. UAVs could fly around while utilizing their sensors to collect all kinds of data as well as record their surroundings with built-in cameras and microphones. Several RL techniques have been proposed to detect objects based on their RF signals. Hence, this article aims to deliver a solution for detecting and identifying intruding UAVs by applying RL techniques to analyze RF signals using the libraries of Python 3 programming language including PyTorch and scikit-learn libraries.

Hierarchical reinforcement learning (HRL) is a technique that decomposes complex tasks into a hierarchy of subtasks. Figure 7 illustrates the HRL architecture where the agent learns distinct policies for each level of the hierarchy through training. Higher-level policies are focused on the planning and coordination, controlling the selection and execution of lower-level policies. Hence, HRL works on organizing and structuring the learning process while enhancing the decision-making process because, at each level of the hierarchy, the designated policy makes informed decisions based on its area of expertise.

Utilizing a hierarchical approach simplifies the learning of the problem, reduces the dimensionality of the action space, and enables the agent to generalize well against unseen data. Breaking the task into smaller subtasks allows for an accelerated learning process by enabling efficient knowledge acquisition and transfer. Furthermore, HRL provides robustness and flexibility to changes in the task specifications. Hence, the agent can adjust specific policies without the need to change the entire framework. Each policy in the agent can be trained with a smaller number of samples compared to non-hierarchical approaches. Overall, through decomposing, HRL allows the agent to gain a deeper understanding of the problem, leading to more effective and efficient learning.

#### 4.2.1. Environment and Elements

In our scenario, we constructed an environment that consists of 10 conceptual channels. Each channel represents a unit within the medium. The state represents an individual channel that holds a single RF data sample. Our system has a UAV agent with multiple policies, each policy is trained for its designated classifier. The agent systematically scans through all channels to inspect the RF activity in each one to determine the appropriate action as a class prediction based on the classifiers of the hierarchy. Granting a reward is dependent on the action taken, the agent will receive a positive reward upon correct predictions. In on-policy algorithms, such as REINFORCE, the policy learns and improves by interacting with the environment. The agent collects experience by following the current policy, and this experience is iteratively used to update the parameters of the policy. Table 7 provides a clear description of the different elements within our scenario.

#### 4.2.2. REINFORCE Algorithm

Our HRL approach trains an agent with multiple policies using a policy gradient method based on the REINFORCE algorithm to solve the problem of UAV detection and identification. In HRL, we use the REINFORCE algorithm proposed by [59]. REINFORCE applies Monte Carlo sampling to estimate the policy gradient and is derived directly from the policy gradient theorem [19]. The policy gradient is a popular approach in the RL field that aims to maximize the reward signal. By formulating the problem as a hierarchical task, we effectively train different policy models to distinguish between UAVs and non-UAVs, as well as classify different models of the UAVs and their flight modes.

After pre-processing the input data using the filtering techniques, our HRL approach trains each policy model such that the policy learns to make decisions specifically for its corresponding classifier. The soft-max function is commonly used to transform the policy’s output into a probability distribution over the available actions. Each action in the policy is assigned a probability, indicating the likelihood of selecting that action given the input features. Ultimately, actions that have higher probabilities are more likely to be chosen, but to encourage exploration, the final selection of the action remains uncertain due to the inherent stochastic nature of the process.

Algorithm 1 outlines the key steps involved in our HRL approach, highlighting the REINFORCE algorithm with an entropy regularization term:

The training of the HRL approach begins at line 4, where a series of episodes start. The agent applies the policy function π(s|θ) to determine the appropriate action based on the current state. Then the state S, selected action A, and reward R are saved as a trajectory. At the end of each episode, the saved trajectories in line 5 are used to update the policy parameters $\theta_{t}$ for future episodes by applying the concept of REINFORCE algorithm.
**Algorithm 1** REINFORCE 1:Input α learning rate, γ discounted factor 2:Initialize environment E 3:Initialize policy parameters θ 4:**for** episode in 1 …N **do** 5:  Use $\pi \left(s|\theta \right)\;to\;collect\;|E|\;trajectories:\;S_{0},\;A_{0},\;R_{0},\ldots ,\;R_{T}$ 6:  G=0 7:  **for** t = T−1 … 0 **do** 8:    $G=R_{t}+\gamma G$ 9:    Compute entropy regularization $ER_{t}=-\sum_{\alpha }\pi \left(A_{t}|S_{t}\right)Log\pi \left(A_{t}|S_{t}\right)$10:    $\hat{J(\theta_{t})}=\gamma^{t}GLog\pi \left(A_{t}|S_{t},\theta_{t}\right)-ER_{t}$11:    $\theta_{t+1}=\;\theta_{t}+\;\alpha \nabla \hat{J(\theta_{t})}$12:  **end for**13:**end for**

For each trajectory t in the list of trajectories, we calculate G as indicated in line 8: (1)$$G=R_{t}+\gamma G$$

*G* represents the cumulative return across a consecutive series of time steps in which it captures the future expected sum of rewards. In each iteration, the immediate reward $R_{t}$ is added to the product of the discount factor (gamma γ) and the previous cumulative return which results in γG, this is to guide the policy of the agent throughout the learning and decision-making process. Utilizing the concept of discounted rewards proves beneficial due to its ability to mitigate the high variance problem that can happen in the REINFORCE algorithm. By including the future rewards until the end of the episode, the concept of Monte Carlo is applied in the REINFORCE algorithm.

After that, we incorporate entropy regularization term into the policy gradient method to improve the training process by increasing the reward gain and encouraging more exploration [19,60] as seen in line 9:(2)$$ER_{t}=-\sum_{\alpha }\pi \left(A_{t}|S_{t}\right)Log\pi \left(A_{t}|S_{t}\right)$$

By introducing entropy regularization term, we aim to make the policy generalize well to unseen data. $ER_{t}$ is computed based on the probability distribution over actions $\pi (A_{t}|S_{t})$ determined by the state $S_{t}$.

In every iteration, we compute the discount factor (gamma γ) for each time step $\gamma^{t}$, which allows for time-dependent discounting of future rewards. Then the objective function of the policy gradient $\hat{J(\theta_{t})}$ is computed based on the equation in line 10. The objective function considers the probability distribution over actions $\pi (A_{t}|S_{t},\theta_{t})$ given the current state $S_{t}$ and the policy parameters $\theta_{t}$. The primary goal of the objective function $\hat{J(\theta_{t})}$ is to find the optimum policy parameters that result in the highest possible long-term reward.
(3)$$\hat{J(\theta_{t})}=\gamma^{t}GLog\pi \left(A_{t}|S_{t},\theta_{t}\right)-ER_{t}$$

To maximize the performance of the algorithm, the policy parameters $\theta_{t}$ are updated based on the observed rewards and the gradients of the policy. We use the equation in line 11 for updating the policy parameters $\theta_{t+1}$: (4)$$\theta_{t+1}=\;\theta_{t}+\;\alpha \nabla \hat{J(\theta_{t})}$$

The term $\alpha \nabla \hat{J(\theta_{t})}$ represents the stochastic estimate that captures the probabilistic nature of the gradient approximation where α is the learning rate of the policy. $\alpha \nabla \hat{J(\theta_{t})}$ is used to measure the performance considering its argument $\theta_{t}$.

To better understand the key components and their interactions as well as the underlying dynamics of the REINFORCE algorithm, Figure 8 presents a visual representation illustrating the flow and organization of the training process using the REINFORCE algorithm.

#### 4.2.3. Hierarchy Design

The droneRF dataset is built using three experiment levels with different numbers of classes in each level. The first level determines the presence of the intruding UAV in the environment leading to having a binary classification problem (‘UAV’ or ‘No UAV’). On the other levels, however, there are more than two classes, which makes them multi-class classification problems. The hierarchical ML approach [24] defined a four-classifier hierarchy that we adopted into our solution to enhance the classification results. The classifiers implemented into the hierarchy are the following:**Classifier 1 (binary classification)**: Determines the UAV presence in the environment, (UAV or NO UAV).**Classifier 2 (multi-class classification)**: Contains three classes (Parrot Bebop, Parrot AR, Phantom). Determines the model of UAV after detecting its existence through the first classifier.**Classifier 3 (multi-class classification)**: Contains four classes (ON, hovering, flying, flying and recording videos). Determines the flight mode of the Parrot Bebop model after passing through the first and second classifiers.**Classifier 4 (multi-class classification)**: Contains four classes (ON, hovering, flying, flying and recording videos). Determines the flight mode of the Parrot AR model after passing through the first and second classifiers.

Figure 9 illustrates the design of the HRL approach. the process begins when the binary classifier ‘Classifier 1’ receives the smoothed and filtered RF sample. The outcome of ‘Classifier 1’ determines the presence of UAV in the environment. If no RF activity is detected through that sample, then the detection process ends, and the outcome is class-1. On the other hand, if the classifier confirms the existence of a UAV in the air, the RF sample goes through ‘Classifier 2’ to determine the UAV’s making model: Parrot Bebop, Parrot AR, or Phantom 3. If the classifier detects a UAV of the model Phantom 3, then the identification process stops, and the outcome is class-10. Based on the outcome of ‘Classifier 2’, the sample is then channeled to either ‘Classifier 3’ (if the detected model is Parrot Bebop) or ‘Classifier 4’ (if the detected model is Parrot AR). Both ‘Classifier 3’ and ‘Classifier 4’ consist of four different UAV flight modes: ON, hovering, flying, and recording videos. If the detected sample goes through ‘Classifier 3’, then the classes for the flight mode of a Parrot Bebop model are class-2 for ON, class-3 for Hovering, class-4 for Flying, and class-5 for Recording Videos. Subsequently, if the detected sample goes through ‘Classifier 4’, then the classes for the flight mode of a Parrot AR model are class-6 for ON, class-7 for Hovering, class-8 for Flying, and class-9 for Recording Videos.

#### 4.2.4. Hierarchical Reinforcement Learning (HRL) Approach Algorithm

The HRL approach applies the hierarchical structure and the REINFORCE algorithm to detect and identify intruding UAVs within the constructed environment. This section presents the pseudocode for the overall HRL procedure, outlined in Algorithm 2, which provides a step-by-step description of the detection and identification process.
**Algorithm 2** Hierarchical Reinforcement Learning (HRL) 1:Load droneRF dataset 2:Define input features and target variables 3:Apply filtering and smoothing techniques 4:Encode the target with respect to the input features 5:Split the dataset into training (70%) and testing (30%) sets (*trainSet*, *testSet*) 6:Create an instance from the environment class (*Env*) 7:Create an instance from the Agent class (agent: Policy 1, Policy 2, Policy 3, Policy 4) 8:Train each policy on *trainset* using *Hierarchical RL* procedure 9:Test each policy on *testSet* and evaluate the system10:**procedure Hierarchical_RL(dataset, agent):**11:**for** episode in 1…N **do**12:  Reset *Env* variable to its original state13:  **for** channel in 1…10 **do**14:    Sample will pass ‘Classifier 1’ to determine the presence of UAV (2 classes: 0-No UAV, 1-UAV)15:    Generate action using *Policy 1* of the agent.16:    **if** action == 0 **then**17:     End and save the predicted value (predClass = C1)18:    **else**19:     Sample will pass ‘Classifier 2’ to determine the UAV model (3 classes: 0-Bebop, 1-AR, 2-Phantom3)20:     Generate action using *Policy 2* of the agent.21:     **if** action == 2 **then**22:       End and save predicted value: Phantom3 UAV (predClass = C10)23:     **else if** action == 0 **then**24:       Sample will pass ‘Classifier 3’ to determine the mode of the Parrot Bebop (4 classes: 0-ON (C2), 1-Hovering (C3), 2-Flying (C4), 3-Recording (C5))25:       Generate action using Policy 3 of the agent.26:       **if** action == 0 **then**27:         End and save predicted value: Bebop, ON mode (predClass = C2)28:       **else if** action == 1 **then**29:         End and save predicted value: Bebop, Hovering mode (predClass = C3)30:       **else if** action == 2 **then**31:         End and save predicted value: Bebop, Flying mode (predClass = C4)32:       **else**33:         End and save predicted value: Bebop, Recording mode (predClass = C5)34:       **end if**35:     **else if** action == 1 **then**36:       Sample will pass ‘Classifier 4’ to determine the mode of the Parrot AR (4 classes: 0-ON (C6), 1-Hovering (C7), 2-Flying (C8), 3-Recording (C9))37:       Generate action using Policy 4 of the agent.38:       **if** action == 0 **then**39:         End and save predicted value: AR, ON mode (predClass = C6)40:       **else if** action == 1 **then**41:         End and save predicted value: AR, Hovering mode (predClass = C7)42:       **else if** action == 2 **then**43:         End and save predicted value: AR, Flying mode (predClass = C8)44:       **else**45:         End and save predicted value: AR, Recording mode (predClass = C9)46:       **end if**47:     **end if**48:    **end if**49:    **if** action == true_label **then**50:      reward = 151:    **end if**52:    Save |E| trajectory: (state S, action A, reward R)53:  **end for**54:  Apply REINFORCE to update the policies of the agent using |E| trajectories55:**end for**56:**end procedure**

## 5. Results and Discussion

The effectiveness of the proposed hierarchical reinforcement learning (HRL) approach to detect and identify intruding UAVs using the features of the captured RF signals is discussed and illustrated in this section. In the testing stage, the gradients of the policy are not updated, and the agents’ actions are generated using the trained policy parameters that were obtained after the training process. First, we review the performance metrics employed to assess the effectiveness of our solution. After that, we analyze the learning process of our solution during the training phase. Subsequently, we evaluate the effectiveness of the HRL approach against unseen data in the testing phase. Finally, we conduct a comparative analysis, comparing our proposed HRL approach against previous research work to highlight its advantages and contributions.

### 5.1. Performance Metrics

In this section, we present the metrics employed to evaluate the effectiveness of our HRL approach. We describe a range of metrics utilized in assessing the performance and quality of our proposed solution. Hence, we gain valuable insights into the strengths and limitations of our work, allowing us to draw meaningful conclusions and enable a comprehensive understanding of the performance. Below are the metrics used in testing our approach:**Cumulative Return**: The sum of all rewards the agent received over a single episode by interacting with the environment. It indicates the agent’s overall performance and success in maximizing long-term rewards by measuring the decision-making capabilities of the trained policy. The quality of the policy improves as the agent earns a higher number of rewards in future episodes.**Policy Loss**: The loss function in our solution is designed to guide the training process of the agent. It assigns greater weights to the actions that resulted in positive rewards while decreasing the weights of the other actions. Hence, the weights are updated iteratively to enhance the generated actions of the agent. By optimizing the loss function, we aim to converge towards an optimal policy that maximizes the cumulative reward over time [61].**Entropy Loss**: Measure the difference between the agent’s predicted probability distribution and the target distribution. Minimizing the cross-entropy loss is key to the learning process as it improves the accuracy of the action probabilities. The cross-entropy loss plays an essential role in refining the decision-making capabilities of the agent. This iterative improvement ultimately leads to an enhanced overall performance of the agent.**Accuracy**: The ratio of correctly classified samples to the total number of samples.
(5)$$Accuracy=\frac{number\;of\;correct\;predictions}{total\;number\;of\;predictions}$$**Precision**: The ratio of correctly classified positive samples to the total number of samples predicted as positive. Precision is important when the cost of false positives is high.
(6)$$Precision=\frac{True\;Positive\;(TP)}{True\;Positive\;\left(TP\right)+False\;Positive\;\left(FP\right)}$$**Recall**: The ratio of correctly classified positive samples to the total number of actual positive samples. Recall is also known as sensitivity or true positive rate. It is an important measure especially in detection systems where the cost of false negatives is high. In the detection systems, we want to minimize the rate of missed positive instances.
(7)$$Recall=\frac{True\;Positive\;(TP)}{True\;Positive\;\left(TP\right)+False\;Negative\left(FN\right)}$$**F1-Score**: The harmonic mean of precision and recall. It integrates the precision and recall into a single metric to gain a good understanding of the model’s performance in terms of both positive predictions and missed positive instances.
(8)$$F1\_Score=2\left(\frac{Precision\;\times \;Recall}{Precision+\;Recall}\right)$$

### 5.2. Training Performance Evaluation

During the training phase, we ran several episodes through the detection and identification tasks to facilitate the exploration of a wide range of states and actions by the agent’s policies, thereby building its experience and enabling convergence toward an optimal policy. In this section, we are going to discuss the training evaluation of the detection task.

In our experimental implementation, we use different hyperparameters for effective learning. The learning rate α is set to 0.0001, controlling the step size during the learning process to perform policy updates. The value of the discount factor (gamma γ) is 0.99, which indicates the importance of future rewards compared to immediate rewards. We use Baseline subtraction method to reduce the variance in REINFORCE algorithm. Additionally, the soft-max function is used to produce probability distribution over the agent’s set of actions. Each action is assigned a probability that represents its likelihood of being selected by the agent. During the learning process, the soft-max function uses a temperature parameter to control the level of randomness in the distribution. The soft-max function has many advantages [19]. First, actions that have higher probabilities are more likely to be chosen, but lower-probability actions still have the chance of being chosen. This ensures that the agent explores different actions and gathers more experience over time. Another advantage is allowing the policy to become deterministic. The last advantage is that soft-max can make a simpler version of the policy, which leads to a faster learning process.

Additionally, we introduce entropy regularization term throughout the training to strike a healthy and balanced ratio of exploration and exploitation. Thus, ensuring our agent explores different classification decisions while still converging towards an optimal policy. This leads to improved performance and better generalization in the classification task. By incorporating the entropy regularization term in the computations of the policy gradient using the REINFORCE algorithm, our HRL solution aims to achieve the accurate and reliable classification of UAVs. Our approach takes into account the diverse nature of the classification and identification problem.

The cumulative return plot depicted in Figure 10 provides a concise visual representation of the agent’s performance and the effectiveness of its learned policy. By observing the trend and trajectory of the plot, the cumulative return starts at a relatively low value at the beginning of the training process. Then, as the agent gains experience and improves its policy, the return keeps rising. There is a sustained rising trend without any noticeable decline in the cumulative return. The plot demonstrates the agent’s ability to achieve better performance by obtaining higher returns over time, highlighting its learning and adaptation capabilities.

Figure 11 illustrates the cross-entropy loss, which serves as a measure of uncertainty within the agent’s action distribution. It initially begins at a value of 0.7, indicating a rather high level of uncertainty in the agent’s decision-making process. It instantly drops and stabilizes near-zero values indicating a progressive reduction in the randomness or uncertainty of the agent’s activities over time. This plot provides valuable insights into the exploration–exploitation trade-off by illustrating the agent’s gradual transition from exploratory behavior to a more deterministic and exploitative behavior.

The total loss of our policy is illustrated in Figure 12, which combines the policy loss with cross-entropy loss. The total loss serves as a key component for updating the policy and refining the decisions during training. At the beginning of the learning process, the overall loss hovers around 4. This indicates that the policy is continuously updating the weights of the actions to optimize its performance and minimize the total loss of the policy. After that, the total loss diminishes and eventually approaches near-zero values. This decreasing trend implies that the policy becomes increasingly refined and aligned with our objectives, ultimately achieving a more optimal state.

### 5.3. Testing Performance Evaluation

In this section, we present a comprehensive analysis of testing our HRL approach which is essential to evaluate its effectiveness and true capabilities. Our HRL technique is tested using various evaluation metrics to verify its ability to generalize and make reliable predictions on unseen data.

The average return per episode is shown in Figure 13 illustrating the agent’s performance in the UAV detection task in terms of cumulative return obtained over multiple episodes. The maximum return per episode in our scenario is 10, in the UAV detection task, the agent achieved an average of 9.97, which reflects its strong performance and high level of success. Figure 14 shows the average reward per episode for the UAV identification task. In this task, the agent achieved an average return of 7.3, which shows its ability to achieve successful outcomes.

Furthermore, we used several testing metrics to mitigate the impact of imbalanced data in the used dataset and provide a more comprehensive evaluation of the agent’s performance. This approach helps to ensure the assessment considers various aspects, such as accuracy, precision, recall, and F1 score [62]. A summary of the results of these metrics is provided in Figure 15 on both the detection and identification tasks. The accuracy achieved by the agent in the detection is 99.7% while in the identification it is 73.4%. The precision of the detection task was calculated to be 99.7% while 73.6% was achieved by the identification task. The recall, also known as the true positive rate, was calculated to be 99.8% for the detection and 73.4% for the identification task, indicating the proportion of actual positive instances that were correctly classified. F1 score, which is a combined measure of precision and recall, was calculated to be 99.8% for the detection and 73.3% for the identification task. In the identification task, the agent achieves relatively lower scores, but the performance is still notable, with reasonably high accuracy, precision, recall, and F1 score. This suggests its ability to identify instances, although with some room for improvement. Overall, the agent showcases strong performance in both tasks, particularly excelling in the detection task with highly accurate and precise classifications.

### 5.4. Comparative Analysis: Previous Research and Proposed Approach

In this section, we present a comparative analysis between our HRL approach and previous works in the field of ML and RL. In [24], ensemble learning classifiers are applied within a hierarchical ML model to detect and identify UAVs. They incorporated pre-processing and feature extraction stages on RF signals. They utilized the same droneRF dataset [32] that we used in our implementation. Also, ref. [44] a semi-supervised learning framework, SDAE-LOF, applied a hierarchical classification model to detect intruding UAVs in the presence of other signals such as Wi-Fi and Bluetooth, using CardRF dataset [45]. Finally, ref. [25] introduced the PRPG algorithm, which uses deep RL and policy gradients to improve 3D object detection based on RF signals by reducing cross-entropy loss.

Our HRL approach primarily focuses on accurately detecting UAVs, achieving an outstanding detection accuracy of 99.7%. HRL offers distinct advantages over traditional ML approaches by capturing complex relationships and decision-making processes in UAV detection and identification. By incorporating RF signals in RL techniques, we offer a unique contribution to the field to improve performance and robustness.

The effectiveness and strength of our HRL are supported by important performance measures presented in Figure 16. Our approach achieves exceptional scores in all metrics, with an accuracy, recall, and F1-score of 99.7%, 99.8%, and 99.8%, respectively. While both our approach and [24] utilize the same dataset, the observed differences in performance can be attributed to our specific utilization of HRL techniques in contrast to the ensemble learning classifiers applied in [24]. Additionally, we outperformed the semi-supervised learning framework [44] that uses a different dataset with more signals such as Bluetooth and Wi-Fi.

Also, we further extend our evaluation by examining the impact of cross-entropy loss on our HRL approach against [24], ref. [25] methodologies in the domain of UAV detection and identification. Lower cross-entropy loss is essential to improve the accuracy of the agent’s action probabilities and it signifies better alignment between the agent’s predictions and the ground truth. Figure 17 illustrates three plots, Figure 17a represents the hierarchical ML methodology [24], Figure 17b represents the PRPG algorithm that uses deep-RL and policy gradients algorithm [25], and Figure 17c represents the cross-entropy loss for our HRL approach. This visual comparison allows us to assess the performance of our HRL approach in terms of cross-entropy loss and its relative advantage over the existing methodologies.

The hierarchical ML model [24] in Figure 17a exhibits a fluctuating cross-entropy loss that ranges between near-zero and 3 where lower cross-entropy means greater confidence in making predictions. It is possible that the complexity of the problem or limitations in the optimization process contributes to this fluctuation. The fluctuation suggests that the agent’s predictions may be inconsistent and less reliable in certain instances.

On the other hand, the PRPG algorithm [25] in Figure 17b initially starts at a relatively high cross-entropy loss of 12.5. However, over 10,000 episodes, the loss gradually decreases to near-zero values. Although the reduction highlights the effectiveness of their work, it is important to highlight that achieving this outcome required a significant number of episodes. The extended training duration may lead to concerns regarding the practicality of the PRPG algorithm in real-world scenarios where computational resources are valuable.

Our HRL approach demonstrates a remarkable performance in terms of cross-entropy loss when compared to the hierarchical ML model [24] and the PRPG algorithm [25]. Figure 17c shows our performance with an initial cross-entropy loss of 0.7, which rapidly converges to near-zero values within around 200 episodes. This rapid convergence stands in contrast to the consistent uncertainty observed in [24] in Figure 17a, highlighting our HRL approach’s successful reduction in the agent’s action uncertainty. Moreover, our approach shows the capability to quickly adapt, learn from data, and optimize the agent’s performance within a shorter training period, unlike [25] represented in Figure 17b, which requires more episodes to achieve comparable levels of convergence.

In addition to evaluating the performance measures discussed earlier, we also analyzed the time efficiency and energy consumption of our HRL approach compared to the hierarchical ML model [24]. The results of time and energy consumption show significant improvements in both aspects. The HRL approach demonstrates an accelerated performance in Figure 18a, with a reduced time of 10.57 s, significantly faster than the hierarchical ML model’s execution time of 19.85 s. This substantial 46.74% decrease in processing time is attributed to the effective decision-making process within our HRL approach, resulting in a faster UAV detection and identification process.

Moreover, our HRL approach demonstrates a great energy efficiency in Figure 18b, consuming only 2.1 joules, which is equivalent to 583.33 micro-watt-hours (µWH), whereas the hierarchical ML approach consumes more than double that amount, with 5.85 joules (1625.0 µWH). Such energy efficiency can be particularly beneficial in scenarios where energy conservation is crucial, enabling longer battery life and reducing overall energy usage. By effectively utilizing energy resources, our approach contributes to more sustainable and cost-effective operations in UAV detection and identification. These results emphasize the positive influence of our HRL approach, demonstrating a significant gain in both time and energy efficiency when compared to the hierarchical ML approach [24].

## 6. Conclusions

The wide utilization of UAVs in various domains has raised concerns about their potential misuse for illegal activities and unauthorized data gathering. As a result, reliable detection and identification systems for intruding UAVs have become a crucial area of research. RL techniques have shown promising results in terms of object detection, employing diverse approaches such as RF, visual, and acoustic-based detection. However, the use of RF-based approaches for the detection of intruding UAVs remains relatively unexplored, making our research a valuable contribution to the field.

Our HRL approach involves using a single UAV agent with multiple policies on a hierarchy of four classifiers to overcome the low accuracy problem caused by non-hierarchical techniques. Each classifier is associated with its corresponding policy. These policies are trained using the REINFORCE algorithm with entropy regularization term. Integrating entropy regularization and adjusting the parameters promotes the policies of the UAV agent to reduce the entropy for a more focused decision-making process. Also, all policies use the soft-max function to enhance the probability distribution over actions, leading to better decision-making in the detection and identification tasks.

Furthermore, we extensively evaluate the performance during training and testing by employing several measures including loss, average return per episode, accuracy, time, and energy consumption. The results demonstrate the excellent performance of our HRL RF-based UAV detection and identification system. HRL successfully detects intruding UAVs with a remarkable detection accuracy of 99.7%. In addition, we observed improvements in the cumulative return while reducing the total policy loss throughout the training which makes it suitable for real-life implementations.

Although our HRL approach demonstrates exceptional performance in drone detection, there are several limitations that require further investigation. The proposed approach is trained using a dataset that is focused on a limited number of UAV types, which in turn limits the variety of UAV signatures and may not cover a full range of possible UAV variations. Additionally, relying on the assumption that the UAVs are exclusively operating within the 2.4 GHz frequency band, introduces the possibility of overlooking UAVs that are utilizing alternative frequency bands. The absence of passive frequency scanning restricts our ability to increase the span of detection to the UAVs that are not actively transmitting RF signals. Addressing these limitations in the future will improve the overall performance and practicality of the HRL approach, especially in the UAV identification task. Thus, contributing to ongoing efforts that ensure the safe and secure use of UAVs across various applications.

## Figures and Tables

**Figure 1 sensors-24-01870-f001:**
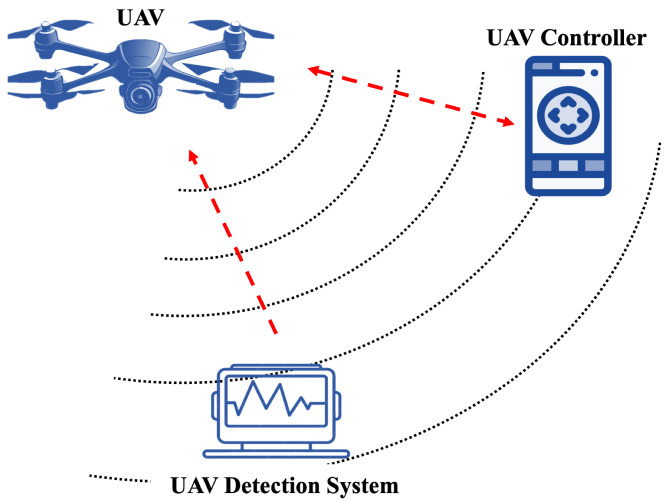
System model for UAV detection: detection of the emitted RF signals from the communication between the UAV and its controller.

**Figure 2 sensors-24-01870-f002:**
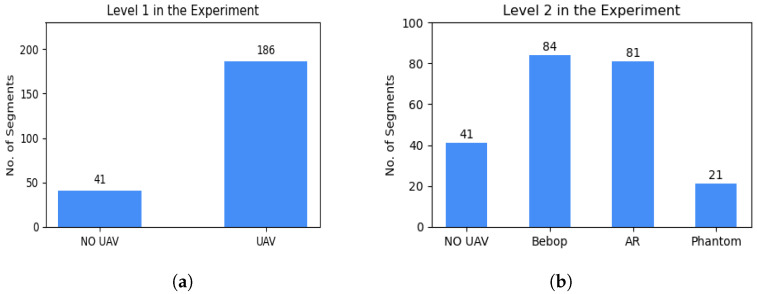
Visual representation of the number of segments in level one and level two of the experiment. (**a**) First level with two classes (UAV, NO UAV). (**b**) Second level represents the presence of the UAV with three classes.

**Figure 3 sensors-24-01870-f003:**
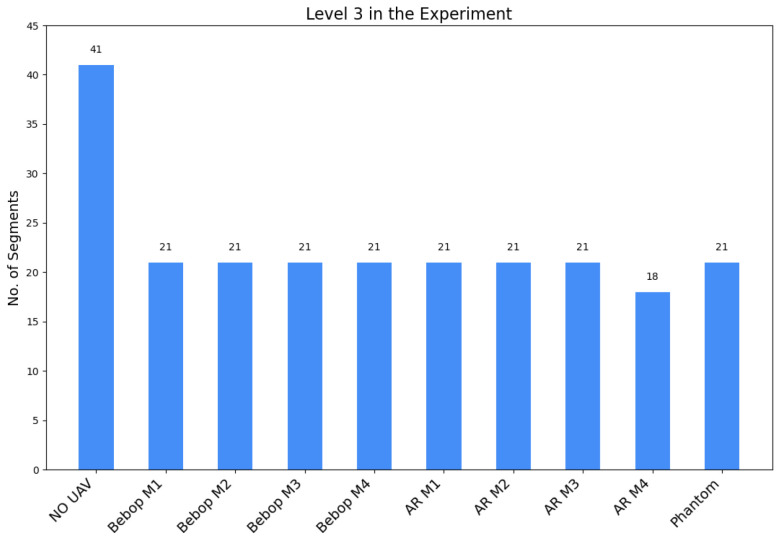
The number of segments in the third level of the experiment. The graph consists of 10 classes, the existence of intruding UAV, its model, and the flying mode.

**Figure 4 sensors-24-01870-f004:**
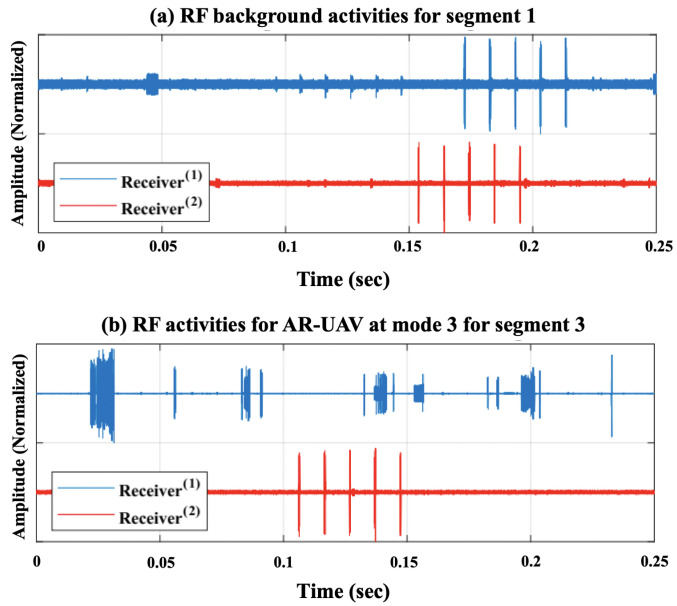
Normalized RF activities (scaled between 1 and −1) (**a**) segment 1 representing no RF activity (**b**) segment 3 representing a flying UAV of the model AR.

**Figure 5 sensors-24-01870-f005:**
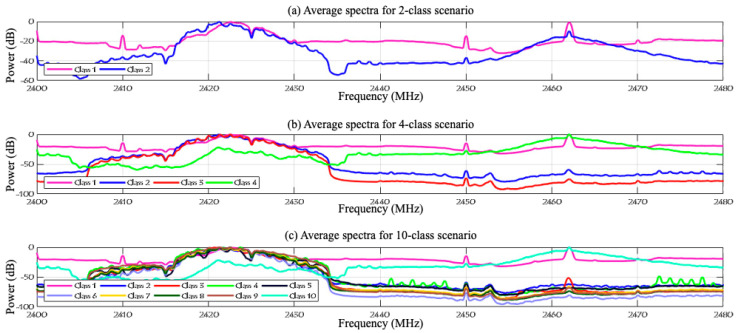
The average spectrums of the three experiment levels.

**Figure 6 sensors-24-01870-f006:**
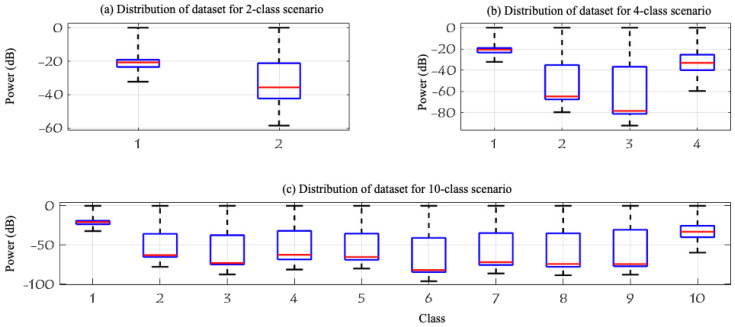
Data distribution of the three experiment levels illustrating the power of each class in the dataset.

**Figure 7 sensors-24-01870-f007:**
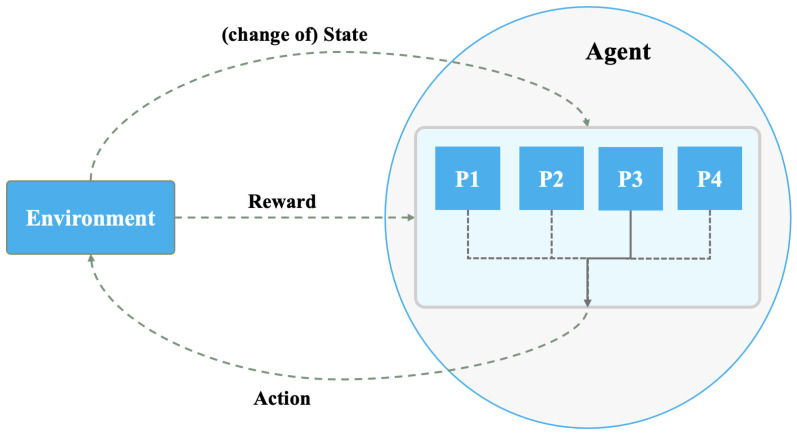
Hierarchical reinforcement learning (HRL) architecture.

**Figure 8 sensors-24-01870-f008:**
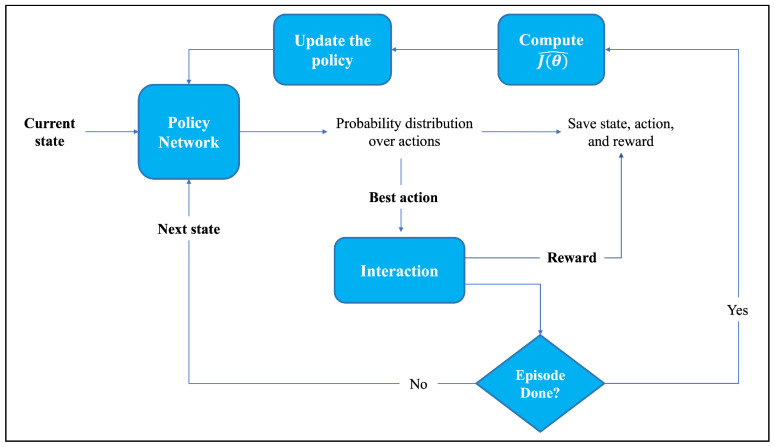
Overview of the training architecture using the REINFORCE algorithm.

**Figure 9 sensors-24-01870-f009:**
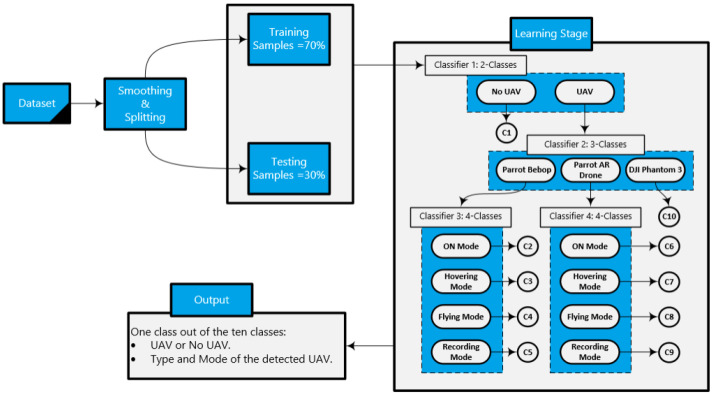
The hierarchical learning approach has two stages: splitting data into 2 subsets, a training set and testing-set, then a learning stage that uses four levels in the hierarchy to identify the class of a given sample [24].

**Figure 10 sensors-24-01870-f010:**
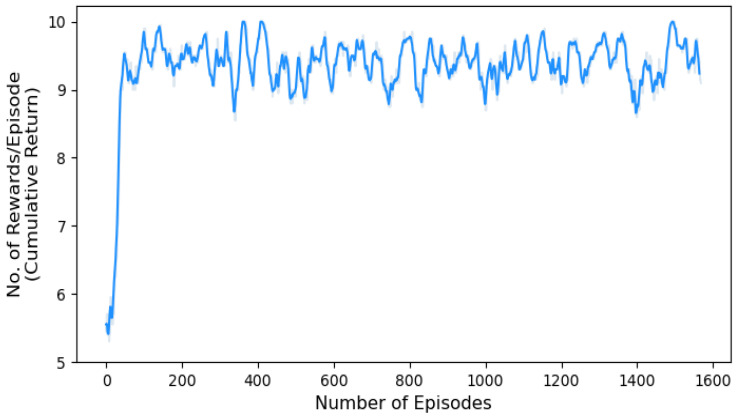
Convergence of cumulative return, representing gained rewards per episode throughout the training of the agent.

**Figure 11 sensors-24-01870-f011:**
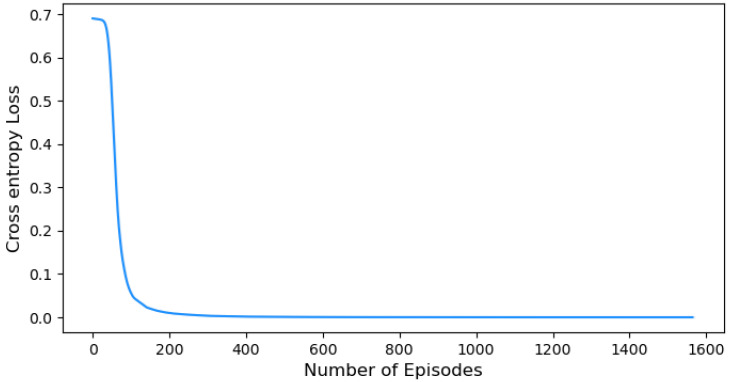
Convergence of cross-entropy loss during the training process.

**Figure 12 sensors-24-01870-f012:**
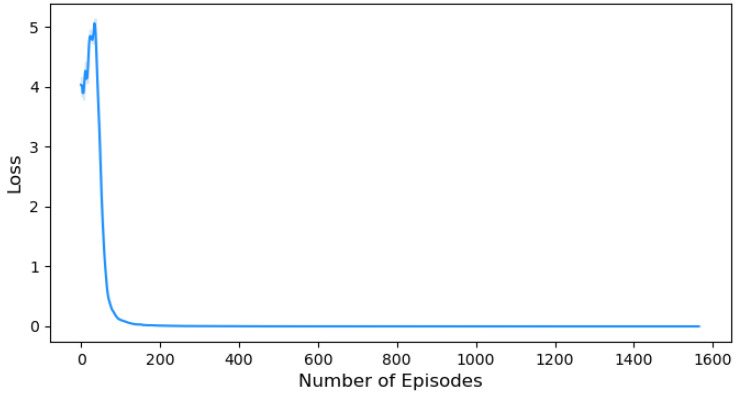
Convergence of total policy loss during the training process.

**Figure 13 sensors-24-01870-f013:**
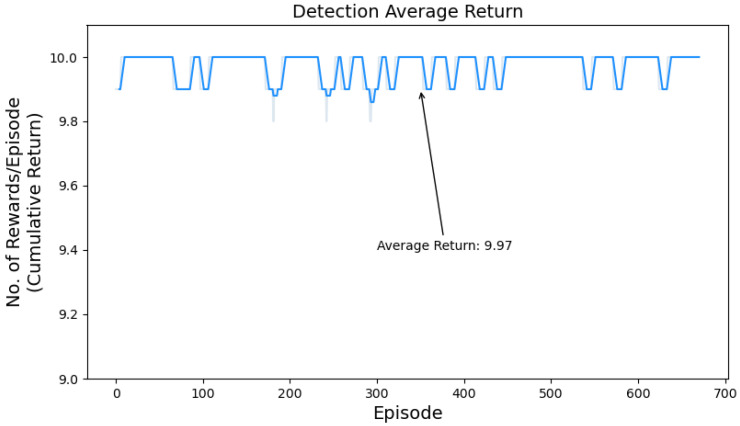
Average return per episode for binary classification in the detection task.

**Figure 14 sensors-24-01870-f014:**
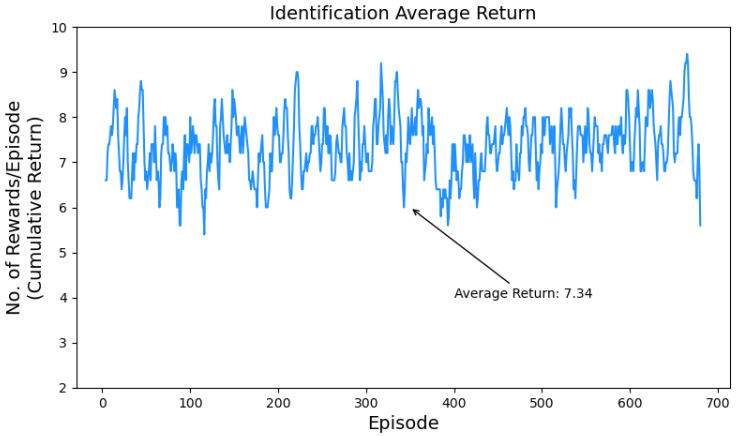
Average return per episode for 10-classes classification in the identification task.

**Figure 15 sensors-24-01870-f015:**
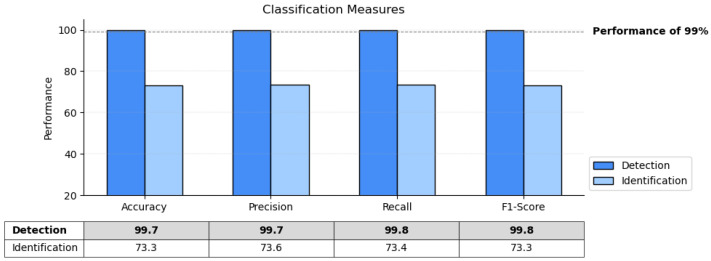
Performance evaluation of agent in detection and identification tasks: accuracy, precision, recall, and f1-score metrics.

**Figure 16 sensors-24-01870-f016:**
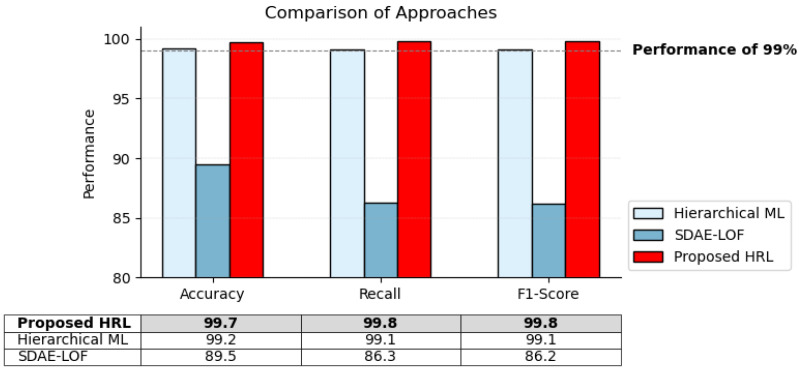
Comparison of accuracy, recall, and F1 score: proposed HRL vs. hierarchical ML and SDAE-LOF.

**Figure 17 sensors-24-01870-f017:**
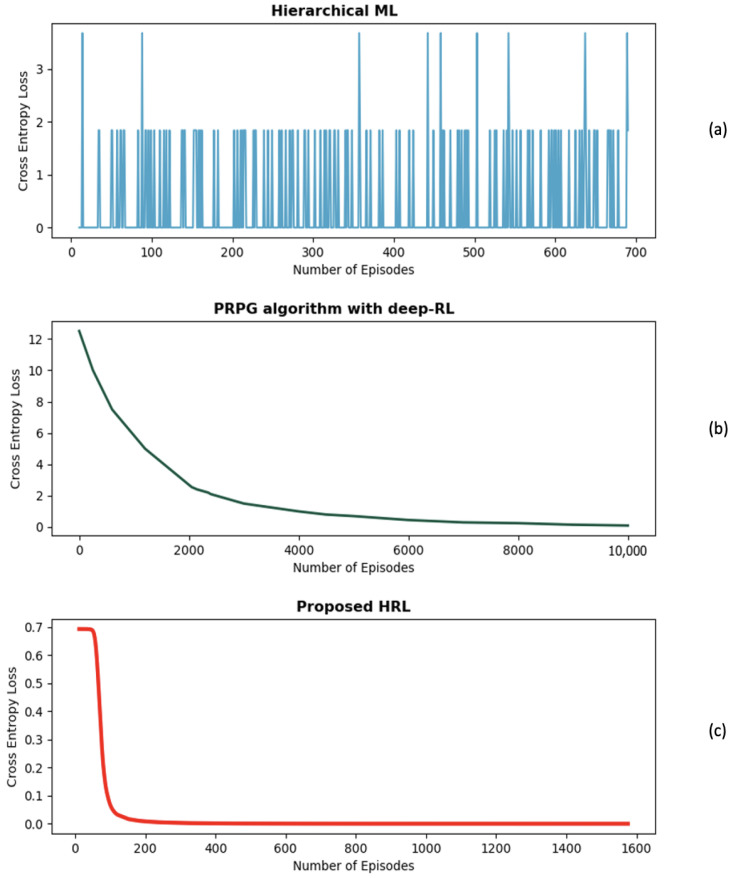
Comparison of cross-entropy loss: (**a**) proposed HRL vs. (**b**) hierarchical ML and (**c**) PRPG algorithm with deep-RL.

**Figure 18 sensors-24-01870-f018:**
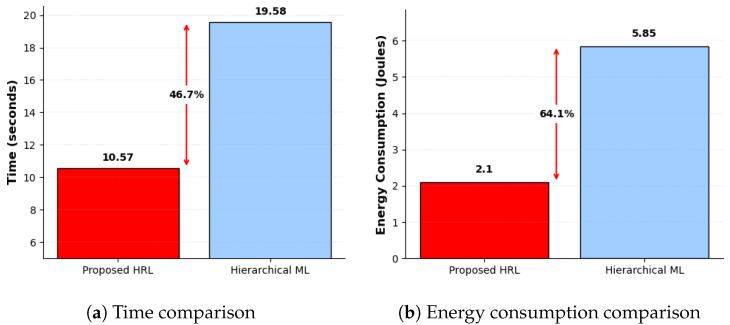
Comparison between the proposed HRL and Hierarchical ML [24] to ensure efficient resource management by conserving the energy and time.

**Table 1 sensors-24-01870-t001:** Summary of UAV detection approaches based on their RF using machine learning.

Ref.	Dataset	Training Model	Measure	Percentage
[33]	Signals collected using a receiver and stored in the computer	ANN	Accuracy STD	>80%
[34]	5000K samples for every controller that contains 100 RF signals	kNN, SVM, DA, and NN	Accuracy False Alarm	99.8% 2.8%
[35]	100 RF signals database. 80% for training. 20% for testing	kNN, SVM, DA, and NN	Accuracy	96.2%
[26]	Windows of dimension 200×3276×2	CNN	MAE	$3.67^{\circ }$
[30]	227 segments where each segment is $20\times 10^{6}$ samples	Three DNN	Accuracy	2 classes→ 99.7% 4 classes → 84.5% 10 classes → 46.8%
[24]	227 segments where each segment is $20\times 10^{6}$ samples	Ensemble Learning	Accuracy	99%
[44]	Mix of UAVs, Wi-Fi, and Bluetooth signals	SDAE-LOF and XGBoost	Accuracy	89.5%

**Table 2 sensors-24-01870-t002:** Summary of visual and acoustic object detection approaches using RL techniques.

Ref.	Feature	2D/3D	Training Model	Measure	Percentage
[46]	Images	2D	The UAV’s alignment and chasing pace to track the intruding UAV	Closest to the target and keep sight of the target	Total reward Absolute value error
[47]	Images	2D	Minimize response time while maximizing the performance	Detect and catch intruding UAVs	Response time Success rate
[23]	Images	2D	The best refined target localization while maximizing the performance	Shortest path distance	Average precision
[22]	Images	2D	Minimize searching time	Target coverage	Average precision
[48]	Acoustic	3D	Enhance localization accuracy and minimize energy consumption	Better localization and lower energy usage	Localization RMSE Energy consumption

**Table 3 sensors-24-01870-t003:** Summary of radiation source detection approaches using RL techniques in a 2D environment.

Ref.	Objectives	Reward	Measure
[49]	Minimize searching time	Closest to the target	Average searching time Failure rate
[50]	Maximize the information gain and reduce the searching time	Obstacle avoidance and shortest path distance	Running time Success rate
[51]	Shortest path to the source	Shortest path distance	Median Completion rate

**Table 4 sensors-24-01870-t004:** Summary of RF object detection approaches using RL techniques in a 3D environment.

Ref.	Objectives	Reward	Measure
[21]	Faster target detection and mapping	Closest to the target	Time Success rate
[25]	Maximize accuracy and minimize cross entropy	The negative cross entropy loss	Speed Cross entropy loss
[55]	Faster performance and avoiding obstacles	$RSS_{t}-RSS_{t-1}$	Speed
[56]	Maximize the accuracy of localization	Belief dependent reward	Inference speed Localization error
[57]	Occupancy detection in indoor areas	Classifying occupied room	F1-score

**Table 5 sensors-24-01870-t005:** UAV specifications based on their model.

UAV	Parrot Bebop	Parrot AR	DJI Phantom 3
Dimensions (cm)	38×33×3.6	61×61×12.7	52×49×29
Weight (kg)	0.4	0.420	1.216
Battery capacity (mAh)	1200	1000	4480
Maximum Range (m)	250	50	1000
Connectivity	WiFi (2.4 & 5 GHz)	WiFi (2.4 GHz)	WiFi ([2.4–2.483] GHz) RF ([5.725–5.825] GHz)

**Table 6 sensors-24-01870-t006:** Specifications of USRP-2943 RF receiver used to create droneRF dataset.

Specifications	Values
Number of channels	2
Frequency Range	[1.2–6] GHz
Frequency step	<1 KHz
Gain range	[0–37.5] dB
Max instantaneous bandwidth	40 MHz
Max I/Q sample rate	200 MS/s
ADC resolution	14 bits
Noise Figure	5 dB to 7 dB
sFDR	88 dB

**Table 7 sensors-24-01870-t007:** Description of the environment and other elements in our proposed HRL solution.

Elements	Description
Environment	10 conceptual RF analysis channels
State	Single channel holding a single data sample
Agent	Single agent for each classifier of the hierarchy
Action	Class number determined by probability distribution
Reward	Positive reward for correct classifications, 0 otherwise

## Data Availability

The data presented in this study are openly available in the Mendeley Data repository. This data were accessed on 5 February 2023 and can be found here: https://data.mendeley.com/datasets/f4c2b4n755/1, reference number [32].

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
