# Peer review of "UAV Detection Using Reinforcement Learning"

_sensors, 2024, doi:10.3390/s24061870_

Round 1

Reviewer 1 Report

Comments and Suggestions for Authors

This article  proposes a new method for detecting and identifying intruding UAVs based on their radio frequency (RF) signals, using a hierarchical reinforcement learning technique. This approach trains a UAV agent with multiple policies using the REINFORCE algorithm with entropy regularization to improve accuracy. By focusing on extracting features from RF signals, the method achieves a remarkable detection accuracy of 99.7%. The research contributes to the field of reinforcement learning by exploring a less commonly used UAV detection approach and enhances UAV security and surveillance capabilities. However, I have some comments that must be considered for this paper to be published.

- The introduction and study of the state of the art are excellent, they are complete, and updated, and comparative tables are presented.

- The architecture of the methodology presented in Figure 1 is incomplete and lacks detail. The architecture must clearly show each of the stages of the proposed architecture in block diagrams, as well as the components of the reinforcement learning system. I recommend breaking down the 3 blocks that are currently indicated and showing the stages that make up each of these in the figure.

In line 515: "This stage involves performing a filtering process to further remove noise and interference." What kind of filter was used?? In addition, it is commented that "we employ methods to enhance data quality and extract meaningful features". What methods were used and why?? I suggest showing the mathematical equations of each filter used.

In line 527: "This stage employs a uniform filter with a predefined window size, utilizing an averaging concept." What is the equation of the filter? What is the selected window size, and why do you use it??

- Why do you use Hierarchical reinforcement learning (HRL) and not other RL methods??

- The agent structure and learning method, as well as the rewards, the actions, and the states (which I think are observations in this case), should be clearly defined in this work. This is key to better understand this approach.

- Line 568: " The agent will receive a positive reward upon correct

predictions". Does this mean you have the ground truth for this signal?? if you have the ground truth, why not just use a supervised learning method??? I recommend comparing the proposed approach with supervised learning methods to analyze metrics such as performance, and required training time, among others.

- You mention using The policy gradient algorithm, but this algorithm is efficient with continuous actions. You mention in the article that you have different discrete categories that you want to recognize (classification problem), is this correct?? Then why use policy gradient?? It is not clear to me what the states, actions, rewards, and observations are for this article. This is key to understanding the method used.

- I strongly suggest to add a discussion section.

Reviewer 2 Report

Comments and Suggestions for Authors

In this article, the authors propose a novel approach for detecting and identifying intruding UAVs based on their RF signals by using a hierarchical reinforcement learning technique and train a UAV agent hierarchically with multiple policies using the REINFORCE algorithm with entropy regularization term to improve the overall accuracy. The research focuses on utilizing extracted features from RF signals to detect intruding UAVs, which contributes to the field of reinforcement learning by investigating a less explored UAV detection approach. However, there are following comments:

1.The authors are requested to provide all the tabular form of all the abbreviations provided in the paper.

2.What are the limitations of the proposed method.

3.Provide more details about the future work.

4. The method proposed by the author should provide pseudocode to make it more comprehensive, otherwise scattered text or formulas cannot fully demonstrate the process of the method.

5.The author should discuss or comment more on the research in related fields or methods in the past three years to highlight the innovation of their research work, i.e.,

Wu J, Li P, Bao J, et al. Quick multiband spectrum sensing for delay-constraint cognitive UAV networks[J]. IEEE Sensors Journal, 2022, 22(19): 19088-19100.

Comments on the Quality of English Language

The language of the paper is fluent, making it easier for readers to grasp the core of the paper.

Reviewer 3 Report

Comments and Suggestions for Authors

I want to thank the authors for their work on the subject.

Author Response

Thank you

Reviewer 4 Report

Comments and Suggestions for Authors

Congratulations to the authors on a truly outstanding article!

Author Response

Thank you

Round 2

Reviewer 1 Report

Comments and Suggestions for Authors

I thank the authors of this article for considering all the recommendations made in the previous review. In particular, the explanation of how the environment, agent, rewards, and states are constructed to be able to use the RL algorithm in this application has been substantially improved. Likewise, substantial improvements have been noted in the conclusions part of the article, and a section has been added with all the abbreviations of this work. In this state, this paper can be published without problem.